# The Role of the Fibronectin Synergy Site for Skin Wound Healing

**DOI:** 10.3390/cells11132100

**Published:** 2022-07-02

**Authors:** Irene Gimeno-LLuch, María Benito-Jardón, Gemma Guerrero-Barberà, Natalia Burday, Mercedes Costell

**Affiliations:** 1Department of Biochemistry and Molecular Biology, Universitat de València, 46110 Burjassot, Spain; irene.gimeno@uv.es (I.G.-L.); gemma.guerrero@uv.es (G.G.-B.); natalia.burday@uv.es (N.B.); 2Institut Universitari de Biotecnologia i Biomedicina, Universitat de València, 46110 Burjassot, Spain

**Keywords:** fibronectin, integrin, fibrosis, wound healing, keratinocyte, catch bond, cell migration, myofibroblast, TGF-β1

## Abstract

Skin is constantly exposed to injuries that are repaired with different outcomes, either regeneration or scarring. Scars result from fibrotic processes modulated by cellular physical forces transmitted by integrins. Fibronectin (FN) is a major component in the provisional matrix assembled to repair skin wounds. FN enables cell adhesion binding of α5β1/αIIbβ3 and αv-class integrins to an RGD-motif. An additional linkage for α5/αIIb is the synergy site located in close proximity to the RGD motif. The mutation to impair the FN synergy region (*Fn1^syn/syn^*) demonstrated that its absence permits complete development. However, only with the additional engagement to the FN synergy site do cells efficiently resist physical forces. To test how the synergy site-mediated adhesion affects the course of wound healing fibrosis, we used a mouse model of skin injury and in-vitro migration studies with keratinocytes and fibroblasts on FN^syn^. The loss of FN synergy site led to normal re-epithelialization caused by two opposing migratory defects of activated keratinocytes and, in the dermis, induced reduced fibrotic responses, with lower contents of myofibroblasts and FN deposition and diminished TGF-β1-mediated cell signalling. We demonstrate that weakened α5β1-mediated traction forces on FN^syn^ cause reduced TGF-β1 release from its latent complex.

## 1. Introduction

The skin covers our body and acts as a barrier, protecting against infections, aggressions and water loss. Upon skin barrier disruption, complex cellular and molecular events are activated to repair the damage and restore skin integrity. In adulthood, the outcome of this process can result in scarring and fibrosis [1,2], whereas complete tissue regeneration is observed in fetal wounds and wounds in lower vertebrates and invertebrates [3,4]. Although there is abundant literature about the factors and mechanisms that determine an endpoint of either scarring/fibrosis or regeneration after tissue injury, the process is still poorly understood [5,6].

Fibronectin (FN) is an extracellular matrix (ECM) protein that is assembled in a cell-driven process into a complex, fibrillar network [7]. The FN fibrils form scaffolds for other ECM proteins, such as collagens and proteoglycans [8,9], and thereby, promote ECM maturation and attachment, mechanosignalling and migration of cells. After cutaneous injury, FN is instantly released and deposited by cells, and represents a major ECM component in all stages of the wound healing process. FN extravasates from injured blood capillaries and provides, together with fibrinogen, the ECM of the forming granulation tissue (GT). In the following hours, this provisional matrix is remodelled and enriched by infiltrated hematopoietic cells and dermal and fascia fibroblasts that migrate to the dermal region of the wound [10,11]. This newly formed fibrotic tissue provides structural support for migrating cell adhesion and constitutes the foundation for either the regeneration of the wounded dermis or the formation of a permanent scar. Simultaneously, FN is a key substrate for keratinocytes to migrate and re-epithelialize the wound gap in the epidermis [12].

FN harbours the major cell binding motif defined by the arginine–glycine–aspartate (RGD) sequence. The RGD motif is located in the 10th type III module (FNIII10) and binds α5β1, αIIbβ3 (exclusive of platelets) and αv-class integrins [13]. In the adjacent module (FNIII9), FN contains the so-called synergy motif that binds α5β1/αIIbβ3 integrins but not αv-class integrins and mediates the formation of catch bonds [14,15,16]. The mutation of the FN synergy sequence in mice underscored its role in resisting or producing high forces, although up to a certain force threshold, αv-class integrins binding to FN compensate for the loss of the synergy site [17].

The course of fibrotic processes, such as adult skin wound healing, depends on several events that are modulated by mechanical signals and, hence, on strong integrin-FN bonds: (i) the population of basal keratinocytes that initiates migration from the wound margins towards the centre of the wound express and activate α5β1 integrins, and form a sheet that advances by sensing the FN-substrate tension and generating force towards it [18]; (ii) the GT is infiltrated by migrating dermal fibroblasts [5,19] that secrete and assemble FN fibrils and fascia fibroblasts that pull their own ECM of FN and collagens [11,20]; (iii) in the GT, a number of fibroblasts convert into myofibroblasts in a process regulated by mechanical forces and TGF-β1 signals [21]; and (iv) further release of TGF-β1 from its latent form bond to the ECM is induced by integrin-mediated force application to the ECM [22].

In the present paper, we investigated the role of FN synergy site during cutaneous wound healing in mice carrying a dysfunctional FN-synergy motif (*Fn1^syn/syn^*; [17]). We demonstrate that *Fn1^syn/syn^* mice show normal wound closure. However, in the initial stage of healing, *Fn1^syn/syn^* wounds form less granulation tissue, with reduced content of myofibroblasts and FN deposition. In-vitro experiments using kidney and dermal fibroblasts derived from *Fn1^+/+^* and *Fn1^syn/syn^* mice, together with plasma FN (pFN) purified from *Fn1^+/+^* and *Fn1^syn/syn^* mice as substrate, revealed that the FN synergy site is important to withstand mechanical tensions required for cell migration and TGF-β1 liberation from the ECMs.

## 2. Materials and Methods

### 2.1. Mouse Strain

The *Fn1^syn/syn^* mutation is expressed constitutively as knock-in in the *Fn1* gene and consists of a substitution of the two arginines (R1374 and R1379) of the mouse synergy motif (DRVPPSRN) in the FN-III9 module with alanines [17]. *Fn1^syn/syn^* mice are born following a mendelian distribution, showing no developmental defects as described. These mutant mice also acquired same size and weight as their wild-type litter mates and were fertile. Mice were housed in a pathogen-free animal facility and fed with standard mouse diet according to European (Directive 2010/63/EU) and Spanish guidelines for the care and safe use of experimental animals. All mouse experiments were performed in accordance with the Government of the Valencian Community (Spain) guidelines with permission references 2016/VSC/PEA/00070 (skin wounds) and 2016/VSC/PEA/00215 (blood draw).

### 2.2. Cell Lines

The keratinocyte cell line used in our migration assays was kindly provided by Reinhard Fässler (Department of Molecular Medicine, Max Planck Institute of Biochemistry, Munich). Primary keratinocytes from *Kind1^fl/fl^* mice were spontaneously immortalized and subcloned [23]. We used a clone that was not transfected with Cre recombinase and, thus, its genotype was wild type. Keratinocytes were grown in keratinocyte-growth medium (KGM, MEM (Merck, Germany), 5 µg/mL Insulin (Merck, Germany), 10 ng/mL EGF (Sigma), 10 µg/mL Transferin (Sigma), 10 µM Phosphoethanolamine (Sigma), 10 µM Ethanolamine (Merck, Germany), 0.36 µg/mL Hydrocortisone (Merck, Germany), 1× Glutamine (Thermo Fisher Scientific, Spain), 100 units per ml penicillin, 100 µg/mL streptomycin (Thermo Fisher Scientific, Spain), 8% Chelated FCS, 45 µM CaCl_2_) on plastic dishes coated with 30 µg/mL collagen I (Purecol^®^ 3 mg/mL, Cell Systems, Germany) and maintained in low-Ca^2+^ concentration (0.45 µM CaCl_2_) to avoid differentiation.

As the *Fn1^syn/syn^* mutation is not lethal we used these mice to generate fibroblasts and dermal fibroblasts. *Fn1*^+/+^ and *Fn1*^syn/syn^ fibroblasts were isolated from mouse kidney and immortalized using SV40 large T antigen [17]. Once immortalized, cells were maintained in DMEM supplemented with 10% FCS and penicillin-streptomycin.

Dermal fibroblasts (DF) were a primary line isolated from *Fn1*^+/+^ and *Fn1*^syn/syn^ mice. Hair from the back skin was removed using clippers and the skin wiped with 70% ethanol. Skin from the back was split off from muscle and fat tissue. The cleaned skin was cut in pieces of 1 cm^2^ and then placed in 6-multiwell dishes to let them stick for 5 min. Once attached, DMEM medium supplemented with 10% FCS and penicillin-streptomycin was added to the explants. Medium was changed every two days. Tissue explants were removed when sufficient cells migrated out from them. DFs were harvested using trypsin, washed and grown in plastic dishes to confluence.

### 2.3. Antibodies

We used the following primary antibodies: anti-FN (1:1,000, Millipore, Burlington MA USA), anti-α-SMA-Cy3 conjugated (1:500, Sigma-Aldrich, St. Louis, MO USA), anti-paxillin (1:300, BD Transduction, Spain), anti-keratin 6 (1:500, Covance, Spain), anti p-Smad2/3 (1:300, Thermo Fisher, Spain), anti-α5 integrin (1:100, Cell Signaling Technology, The Netherlands) and anti-Ki67 (1:200, Abcam, Cambridge, UK). For flow cytometry, we used anti-α5 integrin (1:200 BD Pharmingen, Spain) and anti β3 integrin (1:200 BD Bioscience, Spain). Secondary antibodies: anti-rabbit Alexa Fluor 488 (1:400 Thermo Fisher, Spain) and anti-mouse Alexa Fluor 647 (1:500, Invitrogen). To stain the cytoskeleton, we used Rhodamine Phalloidin (1:500 Thermo Fisher). To stain nuclei, we used either DAPI (1:10,000, Thermo Fisher) or Hoechst (1:10,000, Thermo Fisher).

### 2.4. Wound Healing Assay

Full-thickness wounds (4 per animal) were made on the back of *Fn1^+/+^* and *Fn1^syn/syn^* 10-week-old females (6 mm diameter Biopsy Punch, Stiefel Laboratories, Germany). The back of each mouse was shaved two days before wounding to synchronize the hair follicles stage. On the day of wounding, mice were injected with 0.3 mg/mL Buprecare (0.1 mg/kg body weight) to relieve from pain, and 30 min later anesthetized with 4% Isofluorane. Analgesia was further administrated 4 h after wounding and the following days. Wounds were photographed immediately after wounding (t0) and at indicated time points after wounding. The initial area of the wound was considered 100% opened and to calculate the % of macroscopic wound closure, the area of the wound at each time was calculated and referred to the wound at t0.

### 2.5. Tissue Processing and Histological Analysis

At 4, 7, 9, 15 and 25 days after wounding (daw), mice were sacrificed, wounds dissected and fixed for 48 h in 70% ethanol. Wounds were bisected through following the direction of the hair and each half of the same wound was embedded in paraffin subsequently to a dehydration process. Finally, samples were cut in 4 μm thick sections and stained with Haematoxylin–Eosin (H&E) using standard protocols or immunostained. For tissue immunostaining, sections were permeabilized after paraffin removal with 0.1% Triton-X-100 (Sigma-Aldrich, Spain) for 20 min and then blocked with 3% bovine serum albumin (BSA) in PBS. When necessary, antigen retrieval was performed in citrate buffer (10 mM citric acid, pH 6 and 0.01% Tween 20) by heating in a microwave oven at maximum power for 10 min and then washed with PBS. Sections were incubated overnight with primary antibodies diluted in 3% BSA, 0.01% Triton-X-100 in PBS, washed and incubated with secondary antibodies diluted in blocking solution for 1 h 30 min at room temperature (RT), washed, incubated with either Hoechst or Dapi solution in PBS and mounted with Gelvatol.

### 2.6. Cell Culture and Immunostaining

Cells were starved overnight (o/n) with DMEM or KGM supplemented with 1% FN-depleted FCS and penicillin-streptomycin [17]. On the day of experiment, cells were detached with TrypLE Express (Gibco, Thermo Fisher, Spain) and 7 × 10^5^ cells were seeded on pFN^wt^ or pFN^syn^ coated glass coverslips (25 mm). For experiments, cells were cultured with DMEM or KGM supplemented with 1% FN-depleted FCS and penicillin-streptomycin.

For immunostaining cells were fixed with 4% paraformaldehyde (PFA) in PBS (for paxillin 2% PFA was used) for 10 min at RT, washed of PBS and permeabilized with 0.1% Triton-X-100 in PBS for 10 min at RT. Next, they were washed with PBS and blocked with 3% BSA in PBS for up to 30 min at RT. Then, cells were incubated with primary antibodies in blocking solution overnight, washed with PBS and followed by an incubation with secondary antibody in blocking solution for 1 h and 30 min at RT. After washing the secondary solution, cells were incubated with DAPI solution in PBS for 5 min at RT and mounted with Gelvatol. Images were taken with a Confocal Laser Scanning Microscope Zeiss 780LSM and analysed by ImageJ.

### 2.7. In-Vitro Wound Healing Assays

Single and collective keratinocyte migration assays were performed using Culture-Insert 2-well Self-Insertion (Ibidi, Germany) to create a gap similar to an in-vivo wound. Next, 24-well plates were coated with pFN^wt^ or pFN^syn^ (10 µg/mL solution) for 90 min at 37 °C, excess of protein was washed away with PBS and finally blocked with 3% BSA in PBS for 30 min at RT to avoid unspecific binding. Blocking solution was removed and plastic wells were dried for 10 min before inserting the Ibidi device.

For single-cell migration experiments, keratinocytes were grown overnight in KGM with 8% FN-depleted FCS. Cells were trypsinized using powder trypsin 0.4 g/100 mL in PBS (Gibco, Thermo Fisher, Spain), centrifuged in serum-free DMEM for 5 min at 890 rpm and suspended in 1 mL of KGM with 1% FN-depleted FCS. 7.5 × 10^4^ cells were seeded inside each culture-insert and allowed to adhere for 5 h. Culture inserts were then removed and cells allowed to migrate and fill the gap. Migration was recorded taking one picture every 5 min.

For collective migration experiments, keratinocytes were centrifuged and resuspended into KGM supplemented with 8% FN-depleted FCS and 1.5 mM CaCl_2_ to allow cell–cell adhesions. As such, 4 × 10^4^ cells were seeded per culture-insert well and cell–cell interaction was left to form in an o/n incubation. Culture insert was then removed and cells allowed to migrate into the gap. If using inhibitors, these were added to the keratinocyte FN-depleted medium during the migration process. We used as proliferation inhibitor 40 μM Cytosine β-D-Arabinofuranoside (Sigma Aldrich, Spain) and as cytoskeleton contraction inhibitor 2 μM Blebbistatin (Sigma Aldrich, Spain). Migration was recorded taking pictures every 10 min.

### 2.8. Keratinocyte Migration on PDMS Gels of Different Stiffness

For migration assays on substrates with different stiffness, PDMS gels (Ibidi µ-Dish 35 mm, high ESS) of 1.5 or 15 kPa rigidity were used to seed keratinocytes. Previously, plasma treatment was used to activate PDMS gel hydrophobic surface for 10 min before coating with pFN^wt^ or pFN^syn^ (10 μg/mL) for 1 h at RT. Consecutive PBS washes were performed to wash out the excess of coating. Cells were then seeded at a concentration of 4 × 10^4^ cells/dish in KGM with 8% FN-depleted serum. Random migration was recorded taking one picture every 15 min.

### 2.9. Scratch Assays

We used *Fn1^+/+^* and *Fn1^syn/syn^* fibroblasts in scratch assays. Initially, fibroblasts were trypsinized and starved overnight with DMEM supplemented with 9% serum replacement medium (SRM) (6.5% AIM-V, 5% RPMI and 1% Non-Essential amino acid Solution) and 1% FN-depleted FCS. On the day of the assay glass coverslips were coated with laminin (10 μg/mL) for 1 h at 37 °C. Excess of laminin was removed with 3 washes of PBS and the coated glass surface was blocked with 3% BSA in PBS for 30 min at RT. Cells were detached with TrypLE, centrifuged and seeded at 3.5 × 10^5^ cells/coverslip in SRM. The confluent culture was scratched with a yellow tip and cell migration towards the gap was recorded over 20 h. After this period, cells were fixed and FN was immunostained to observe the fibrils of the matrix assembled by migrating cells. FN ECM quantification was done as previously described [24,25] using Fiji ImageJ. Briefly, to measure the FN fibrils, the immunostained FN area was measured and normalized to the total area and cell number in each image. To quantify the number of FN branches, images were set to binary and same threshold was applied to all of them, followed by despeckle function to remove background and skeleton plugin. The amount of FN fibres and branches was normalized to the image area.

### 2.10. Cell Migration Analysis

Migrations were recorded either with a GE Healthcare IN CELL Analyzer 2000 (GE Health Care, USA) or with a Zeiss Axiovert microscope (Germany) and the 5.0.0.8 version VisiView (Visitron System, Puchheim, Germany) software. In the analysis of keratinocyte migrations, the Manual Tracking plug-in was used to record cell paths. Frames from 4 to 6 different areas of the open gap created by the insert or scratch were photographed per condition. From each biological replicate, between 6 and 10 different cells were tracked from every recorded area. Information was then loaded into ImageJ Chemotaxis tool. This plugin was used to calculate the average speed; directionality as the ratio distance Euclidean/distance accumulated (De/Da); and distance for all cells tracked in each area. For statistical analysis we performed 3–4 biological replicates for each condition.

### 2.11. Dermal Fibroblasts Conversion into Myofibroblasts on Compliant Substrates

For DF conversion assays, commercial polyacrylamide (PA) gels were used (Matrigen, 35 mm dish, 20 mm glass bottom, soft view, easy-coating hydrogels with 4 and 50 kPa). DFs were starved for 24 h in DMEM supplemented with 1% FN-depleted FCS. Cells were then detached with TrypLE and seeded onto the PA gels previously coated with 10 µg/mL pFN^wt^ or pFN^syn^ for 90 min at RT and blocked with 3% BSA for 30 min. Cells were incubated for 24 h to compliant substrates and then fixed. For each biological replicate, DFs were extracted from different *Fn1^+/+^* and *Fn1^syn/syn^* mice. About 15 pictures were analysed per replicate and statistical significance per each rigidity was calculated from the three different replicates.

### 2.12. TGF-β Bioassay

Transformed mink lung epithelial cells (MLECs), kindly provided by Daniel Rifkin’s laboratory, were maintained in DMEM supplemented with 10% FCS and penicillin-streptomycin. Cells were transfected with an expression vector consisting of the plasminogen activator inhibitor-1 (PAI-1) promotor and fused to the luciferase reporter [26]. MLECs were used to quantify active TGF-β released by *Fn1^+/+^* and *Fn1^syn/syn^* DFs to the culture media. Wild-type or mutant DFs were seeded onto 96-well plates (5 × 10^3^ cells/well) previously coated with 10 µg/mL pFN^wt^ or pFN^syn^. Cells in DMEM with 1% FN-depleted FCS were allowed to attach and form an ECM for 3 days. After 3 days, conditioned media by DFs was collected and added to 96-well plate containing MLECs (2.5 × 10^4^ cells/well). In parallel, MLECs were grown in 0.1% BSA serum-free medium for 4 h. MLECs were then incubated for further 14 h in DF-conditioned media. After the incubation, cells were lysed and the luciferase activity was evaluated with the Luciferase Assay System Kit (Promega) and measured with a luminometer. Levels of total TGF-β1 were assessed by heating DF-conditioned media for 5 min at 80 °C [26]. The amount of free/active TGF-β1 was related to the amount of total TGF-β1 obtained from the same conditioned medium. Statistical significance was calculated with average data coming from 2 to 3 technical replicates per condition.

### 2.13. Flow Cytometry

Cells were detached with trypsin, centrifuged with 10% FCS DMEM medium to inactivate trypsin and resuspended in PBS. Around 5 × 10^5^ cells in suspension were centrifuged and resuspended in 200 µL of FACS buffer (30 mM Tris-base, pH 7.4, 180 mM NaCl, 3.5 mM KCl, supplemented with 1 mM CaCl_2_, 1 mM MgCl_2_, 3% BSA, 0.02% NaN_3_). Same number of cells were distributed in round-bottom 96-well dishes. The 96-well dish was centrifuged, and cells subsequently resuspended in FACS buffer with primary antibody diluted in FACS buffer. Incubation with primary antibodies was performed for 1 h on ice. To remove excess of primary antibody, cells were centrifuged twice and washed with FACS buffer. For non-labelled primary antibodies, cells were resuspended in secondary antibody dilution and incubated 30 min on ice. Unbound antibodies were washed out by two consecutives centrifugations in FACS buffer. Finally, cells were resuspended in 300 µL FACS buffer and analysed with FACS Canto (BD Bioscience, Spain) flow cytometer. Integrin profile was done using 3 technical replicates and compared to non-stained cells and isotype control for each antibody.

### 2.14. Statistical Analysis

All the experiments were conducted using 3–4 independent biological replicates, unless otherwise indicated, and independently processed. Results represented as mean ± standard error of the mean (SEM). Statistical analysis was performed by Student´s *t*-test for independent or paired parametric data, and Mann–Whitney U test was used for independent non-parametric data. A *p*-value less than 0.05 was considered statistically significant.

## 3. Results

### 3.1. Skin Wound Closure in Fn1^syn/syn^ Mice

To evaluate the consequences of the loss of the FN synergy site for skin wound healing, we performed four full-thickness 6 mm diameter excisional wounds (day 0) on the back skin of wild-type (*Fn1*^+/+^) and mutant (*Fn1*^syn/syn^) litter-mate mice (Figure 1A). Macroscopic wound closure was measured at 4, 7, 9, 15 and 25 days after wounding (Figure 1B and Appendix A). At day 4 after wounding, there was a mild difference in the wound size: whereas the wound gap was reduced to approximately 30% of its initial area in *Fn1*^+/+^ mice, the wound gaps were reduced only to 40% of the initial size in *Fn1*^syn/syn^ mice. At day 9 after wounding, almost all wounds of *Fn1*^+/+^ as well as *Fn1*^syn/syn^ mice were macroscopically closed. To further study the potential role of the synergy site in the re-epithelialization process, we used H&E images of wound sections (Figure 1C) to measure the cross-sectional wound lengths in *Fn1*^+/+^ and *Fn1*^syn/syn^ mice (Figure 1D), the mean neo-epidermis thickness (Figure 1E) and the area occupied by the GT at different time points (Figure 1F). At day 7 after wounding, all epidermal layers were closed in both *Fn1*^+/+^ and *Fn1*^syn/syn^ mice (Figure 1C and Appendix A). After fusion of the epidermal tongues, the newly formed epidermis was thinner in *Fn1*^syn/syn^ wounds during the early stages of regeneration (between day 7 and 15 after wounding) but acquired a normal thickness at day 25 after wounding. We measured the wounded dermic area underneath the epidermis during the healing process in H&E-stained sections (Figure 1C,F, indicated with black dotted lines). In *Fn1*^syn/syn^ wounds, GT covered a sectional area of 0.8 ± 0.1 mm^2^ at day 4 after wounding, which had approximately half of the sectional area in *Fn1*^+/+^ wounds (1.4 ± 0.09 mm^2^). In the following days, GT areas were smaller in *Fn1*^syn/syn^ wounds, although the differences were not significant to *Fn1*^+/+^ wounds. Taken together, our data indicate that inactivation of the FN synergy site only mildly impacts the course of wound closure.

### 3.2. Characterization of The Re-Epithelialization Process on FN^syn^

Upon wounding, keratinocytes at the edge of the wound are activated, express certain keratins, such as K6 and K14, and rapidly change their integrin profile. Among them, α5b1 integrins are upregulated in the migrating basal keratinocytes [19]. At day 4 after wounding, the expression of keratins was similar between *Fn1^+/+^* and *Fn1^syn/syn^* newly formed epidermis (Appendix A), while α5 integrin expression was evident in 83% of basal *Fn1^syn/syn^* keratinocytes and only 43% of *Fn1^+/+^* keratinocytes (Figure 2A,B). This increase in α5 integrin-expressing cells may represent a compensatory effect to the compromised adhesion to the mutant FN substrate. To test this hypothesis, we sought to study in vitro the process of keratinocyte migration on FN^wt^ and FN^syn^ -coated substrates. We seeded a mouse wild-type keratinocyte line (see Materials and Methods) that expresses α5β1 [23] but lacks αvβ3 integrin expression (Appendix A) on FN purified from *Fn1^+/+^* and *Fn1^syn/syn^* blood plasma (pFN^wt^ and pFN^syn^) and found similar adhesion, spreading and cell size on pFN^wt^ and pFN^syn^ as well as formation of similar numbers of focal adhesions 5 h after cell seeding (Appendix A–D).

To study single and collective keratinocyte migration, we used 2-well inserts (Culture-Insert 2-well; Ibidi) to create a gap devoid of cells without disturbing the pFN-coated surface. To induce keratinocyte collective migration, we supplemented the keratinocyte culture medium with 1.5 mM CaCl_2_. Single keratinocyte migration on pFN^syn^-coated surfaces was significantly slower compared to keratinocytes migrating on pFN^wt^ (Figure 2C,D). Analysis of cell directionality showed that keratinocytes on pFN^wt^ substrates migrate in a relatively straight direction compared to those on pFN^syn^ substrates, which move more arbitrarily (Figure 2E,F).

Next, we studied single-cell migration of keratinocytes seeded on FN-coated polyacrylamide with different stiffness (1.5 and 15 kPa) (Figure 2G). Lower migration speeds were also observed in keratinocytes migrating on soft substrates coated with pFN^syn^ compared to substrates coated with pFN^wt^. We also analysed keratinocyte collective migration with culture medium supplemented with 1.5 mM CaCl_2_ and tracked the movement of cells in the inner mass and at the sheet edge over a time period of 6 h (Figure 2H–I). Interestingly, a faster movement was observed in keratinocytes migrating on pFN^syn^ in both leading and inner mass compared with cells migrating on pFN^wt^. We excluded that differences between cells on pFN^syn^ and pFN^wt^ were due to different cell proliferation by adding 40 μM Cytosine β-D-Arabinofuranoside to the medium (Appendix A), as the differences remained in the presence of the proliferation inhibitor. Since collective migration is the result of integrin-dependent forces and cadherin-based intercellular forces [27], we evaluated the involvement of cell contractility in our collective migration experiments, treating the cells with the myosin-II inhibitor Blebbistatin. Cell sheets migrating on pFN^wt^ and treated with Blebbistatin increased their velocity, reaching similar values as cells on pFN^syn^ without an inhibitor (Appendix A). Interestingly, the combination of Blebbistatin inhibitor and the synergy site loss in FN decreased the mean speed of the cell sheets to the FN wild-type condition (Appendix A). These results show the importance of the synergy site in force transmission during keratinocyte migration, and that this can be partially compensated by cell–cell interactions during the collective cell sheet advancement.

### 3.3. Formation of the Granulation Tissue in Fn1^syn/syn^ Mice

Myofibroblasts are α-SMA-positive cells that contribute to the formation of the GT by depositing ECM and contracting the wound [28,29]. At day 4 after wounding, the first α-SMA-positive cells appeared in the periphery of the injured area below the epidermis (Figure 3A). In the following days, as the epidermal layer closes, the cloud of myofibroblasts extends to the centre of the wound predominating in the upper half of the dermis. At 7 days after wounding, GT contained the highest levels of myofibroblasts, which progressively decreased in the following days and was practically absent at day 15 after wounding. Although myofibroblast distribution was similar in *Fn1*^syn/syn^ and *Fn1*^+/+^ wounds, at day 4 after wounding, the area occupied by α-SMA-positive cells was significantly reduced in *Fn1*^syn/syn^ when compared to *Fn1*^+/+^ wounds (Figure 3B). Myofibroblasts both secrete high amounts of ECM proteins, including FN, and use FN as a substrate to adhere and migrate [30,31]. At day 4 after wounding, FN appeared in the lateral margins of the GT, co-localizing with α-SMA-positive cells (Figure 3C). At this time, the areas of FN deposits were half the size in *Fn1*^syn/syn^ wounds compared to *Fn1*^+/+^ wounds (Figure 3C,D). In contrast, there was a negligible change in GT area or ECM deposition at later healing stages. Ki67 expression has been used as a dermal proliferation marker upon injury [32]. Then, to further understand the decrease in GT at day 4 after wounding, we studied the presence of Ki67-positive cells in this tissue (Figure 3E,F). As shown, the proportion of Ki67-positive cells was significantly reduced in the GT of *Fn1*^syn/syn^ wounds compared to *Fn1*^+/+^ wounds.

To investigate the function of the FN synergy site in the process of matrix remodelling and fibroblast migration, immortalized *Fn1*^+/+^ and *Fn1*^syn/syn^ fibroblasts isolated from wild-type and mutant mice were seeded on laminin-coated coverslips, and left to secrete and assemble their own ECM in the presence of very low FN-depleted FCS (see Section 2.9). When confluent, the monolayer was scratched with a pipette tip and cells were allowed to secrete and assemble their own substrate to migrate and cover the gap (Figure 4A). We measured the mean velocity of the cells in the migratory front (Figure 4B) and observed a significant reduction in the cell velocity of *Fn1*^syn/syn^ fibroblasts compared to wild-type FN-expressing fibroblasts. *Fn1*^syn/syn^ fibroblasts in the front advanced with slight, although non-significant, lower directionality than *Fn1*^+/+^ fibroblasts (Figure 4C,D). After migration, we stained and characterised FN in the newly assembled ECMs that migrating fibroblasts used as substrate (Figure 4E–G). The FN^syn^ newly formed matrices were significantly less dense than matrices assembled by *Fn1*^+/+^ fibroblasts, whose fibres were shorter and with significantly fewer branches (Figure 4F,G). Altogether, our data suggest that *Fn1*^syn/syn^ fibroblasts assemble FN fibrils more slowly and/or less elaborated than *Fn1*^+/+^ fibroblasts, which consequently limits their migration.

### 3.4. Myofibroblast Conversion Is Reduced in the Absence of the FN Synergy Site

To test whether the reduced numbers of myofibroblasts in the GT of *Fn1*^syn/syn^ wounds were due to an abnormal conversion of dermal fibroblasts to myofibroblasts, we seeded primary *Fn1*^+/+^ and *Fn1*^syn/syn^ dermal fibroblasts (DFs) on PA gels of 4 kPa, 50 kPa and on glass coverslips coated with either pFN^wt^ or pFN^syn^ (Figure 5A). After 24 h in culture, the percentage of α-SMA positive myofibroblasts was calculated. As expected, increased rigidity increased the proportion of *Fn1*^+/+^ myofibroblasts, from 50% (on 4 kPa) to 61% (on 50 kPa). However, in the absence of a functional FN synergy site, the myofibroblast conversion was significantly reduced to 43% (on 4 kPa) and 50% (on 50 kPa), respectively (Figure 5B). On glass, however, differences between genotypes were lost.

TGF-β1-induced cell signalling contributes to fibroblast-to-myofibroblast differentiation [33]. TGF-β1 is secreted as part of the large latent complex (LLC), which, in addition to TGF-β1, consists of latency-associated protein (LAP) and latent TGF-β binding protein 1 (LTBP-1). The LLC provides a reservoir of latent TGF-β1 in the ECM by binding to ECM components, mainly fibrillin-1 and FN [34,35,36,37]. The traction forces produced by myofibroblasts on FN fibrillar networks have been demonstrated to contribute to TGF-β1 release from its latent complex [22]. Thus, we directly tested TGF-β1 release from FN matrices. To that end, we performed a bioassay to measure TGF-β1 in conditioned media by *Fn1*^+/+^ and *Fn1*^syn/syn^ dermal fibroblasts after 3 days of culture. Interestingly, although the levels of free (active) TGF-β1 were mildly reduced in the medium conditioned by *Fn1*^syn/syn^ dermal fibroblasts (Figure 5C), the total amount of TGF-β1 (free form plus assembled into its latent complex form) in the medium derived from *Fn1*^syn/syn^ dermal fibroblasts was significantly increased (Figure 5D). This resulted in a significant decrease in the percentage of active TFG-β1 relative to the total amount in *Fn1*^syn/syn^ (Figure 5E) compared to *Fn1*^+/+^ dermal fibroblast-conditioned medium. Our results suggest that FN^syn^ fibrils trap less TGF-β1-containing latent complex, which may contribute to reducing TFG-β1 release.

TGF-β is commonly considered the master pro-fibrotic cytokine. Upon TGF-β1 binding-induced formation of the tetrameric TβRI/TβRII complex, R-Smads Smad2/3 are phosphorylated and then complex with Smad4 and translocate into the nucleus to mediate transcription of Smad-dependent genes [38]. To confirm our in-vitro data, we analysed TGF-β1-induced signalling in the GT in skin wounds at day 4 after wounding by quantifying the content of phosphorylated-Smad2-positive cells (Figure 5F). The proportion of p-Smad2 positive cells in the GT from *Fn1*^syn/syn^ wounds was reduced compared to *Fn1*^+/+^ wounds (Figure 5G). These findings support the notion that the absence of the FN synergy site impairs TGF-β1 release from the ECM and, hence, impacts the GT formation.

## 4. Discussion

Skin wound healing has been broadly studied in the last decades, as millions of people worldwide require medical care due to impaired healing every year. The new therapies are focused on achieving scarless skin regeneration, with a great focus on animal and human models. Scars are the consequence of fibrotic processes after skin injury, which involve activation of different fibroblast lineages [5], overexpression of ECM components and strong integrin-mediated force transmission. How mechanical signals drive scar formation and/or regeneration still remains an important question.

Upon tissue injury, FN is one of the first and most abundant ECM components deposited into the healing wound [39,40]. Furthermore, cell adhesion to FN triggers biochemical and biophysical signalling pathways, which profoundly determine the outcome of the healing process, leading either to repair or scar formation. Our studies in a mouse model of wound healing revealed that the time when the epidermal tongue closed the wounds was indistinguishable between *Fn1*^+/+^ and *Fn1*^syn/syn^ mice. However, the epidermal thickness was reduced between day 9 and 15 after wounding of *Fn1*^syn/syn^ mice. We observed a normal distribution and differentiation of both basal and suprabasal keratinocytes at the epidermal keratinocyte tongue of *Fn1*^syn/syn^ and *Fn1*^+/+^ wounds, whereas the proportion of α5β1-expressing basal keratinocytes was significantly increased at the epidermal tongue of *Fn1*^syn/syn^ wounds. To better understand re-epithelialization, we studied keratinocyte migration on pFN-coated substrates in vitro. Although in the first stages after wounding, keratinocytes can migrate individually, efficient epidermal tongue movement is produced by collective migration. Therefore, we studied single and collective keratinocyte migration. Interestingly, keratinocytes migrating individually on pFN^syn^ moved slower than keratinocytes on pFN^wt^, both on soft and stiff substrates. The reduced migration velocity of keratinocytes on pFN^syn^ was also associated with reduced directionality when compared to keratinocytes migrating on pFN^wt^. On the contrary, however, collectively migrating sheets of keratinocytes advanced faster on pFN^syn^ than on pFN^wt^. Individual migration requires directional cell polarity involving a leading edge at the cell front and a lagging edge at the rear of the cell. This motion depends on strong traction forces generated by integrin-mediated adhesions: protrusion and adhesion of the leading edge and retraction of the rear edge. It has been shown that cell traction forces are stronger in the perinuclear region than in the cell front. The reason for this disparity is believed to depend on α5β1 integrin-based fibrillar adhesions that retract the rear edge, whereas in the cell front, αvβ3-containing focal adhesions, which produce less force, predominate [41,42]. Our results suggest that the loss of the FN synergy site might limit the forces for cell rear retraction with the consequence of an abnormal migration directionality and reduced velocity. Collective migration, however, depends on a tight balance between cell–cell and cell–ECM bonds [43]. According to our results, weakening of the α5β1 integrin-mediated cell-FN adhesions by the loss of the FN synergy site or by treatment with Blebbistatin increases keratinocyte sheet movement. This observation is in line with reports showing that during collective migration, myosin-II-induced cell rear retraction is compensated by cell–cell adhesions [44]. Our findings also suggest that the absence of the synergy site may diminish the lifetime of FN-α5β1 bonds, and, hence, accelerate binding/unbinding rates and thereby, increase the collective migration speed. The convergence of the two effects, which retarded individual and accelerated collective cell migration, provides an explanation for the apparently normal rate of re-epithelialization of Fn1^syn/syn^ wounds in vivo. It is possible that weak traction forces caused by FN synergy site mutation negatively influence the onset of keratinocyte migration when the epidermal tongue is not yet fully established. This deficit might be compensated by accelerated cell migration in the following days. Moreover, the increased number of α5β1-expressing basal keratinocytes may compensate reduced cell adhesion to FN^syn^.

In the wounded dermis, at day 4 after wounding, *Fn1^syn/syn^* mice form a GT, which was around half the size of the GT in wild-type wounds. The GT of *Fn1^syn/syn^* mice contained less FN and myofibroblasts. Our results are in line with those observed in the fibroblast-specific focal adhesion kinase (FAK) knockout mice [45]. FAK transduces mechanical forces sensed by integrins. An important proportion of the GT fibroblasts is activated to myofibroblasts. In our model, we observed that 4 days after wounding, the number of α-SMA-positive cells in the *Fn1^syn/syn^* GT was reduced to 30% of that observed in *Fn1*^+/+^ GT. We explored putative causes for the α-SMA-positive cell decrease, such as proliferation, fibroblast migration and fibroblast-to-myofibroblast conversion. At 4 days after wounding, the number of proliferative Ki67-positive cells was significantly reduced in the GT of *Fn1^syn/syn^* mice, which could, at least partially, explain the presence of less cells in and the reduced size of the GT. It is possible, however, that other factors, such as migration and differentiation, influence the lower number of α-SMA-positive cells. We studied the motile behaviour of fibroblasts using an in-vitro scratch assay, which revealed that *Fn1^syn/syn^* fibroblasts, which secrete their own mutant FN matrix, displayed a slower cell advance than *Fn1*^+/+^ fibroblasts. In addition, the FN fibrillar matrices assembled by migrating *Fn1^syn/syn^* fibroblasts were less elaborated than FN fibrils assembled by *Fn1*^+/+^ fibroblasts, suggesting that the slower advance of *Fn1^syn/syn^* fibroblast front is possibly due to the impaired assembly of an adequate fibrillar substrate that is also reflected in the content of FN in the *Fn1^syn/syn^* GTs at 4 days after wounding.

Skin rigidity increases along the healing process, ranging from 0.01 to 10 kPa in the fibrin-FN clot, to ~18 kPa in early GT and reaching levels of ~50 kPa in mature GT with increased organization of FN and collagen fibres [46]. The number of α-SMA-expressing fibroblasts at the wound site is determined by a positive feedback loop, in which tension facilitates TGF-β1 release and activation, which, in turn, induces α-SMA expression. Conversion to α-SMA-expressing fibroblasts increases force production and tension development [47,48]. We observed that dermal fibroblasts seeded on surfaces of different rigidity displayed a reduced rigidity response in the absence of a functional FN synergy site. TGF-β1 is activated extracellularly, released from its noncovalently associated LAP protein by different mechanisms, including the action of metalloproteinases and/or cell contraction [35,49]. In the non-proteolytic TGF-β1 activation, integrin-mediated force application on FN fibrils induces conformational changes in LTBP-1 and LAP, which leads to the release and activation of TGF-β1 [22]. We observed that ECMs assembled by *Fn1*^syn/syn^ dermal fibroblasts have a reduced capacity to trap LLC complexes secreted by the same cells. This effect could be, on the one hand, a consequence of the lower complexity of the ECMs assembled by *Fn1*^syn/syn^ cells, which impairs conformations that favour LLC storage. Alternatively, *Fn1*^syn/syn^ dermal fibroblasts could be less efficient to release TFG-β1 due to weakened α5β1-mediated contractile forces on the FN^syn^ (Figure 6). Clearly, the levels of phospho-Smad2-positive cells were reduced in the GT of *Fn1^syn/syn^* wounds, supporting lower TFG-β1-mediated cell signalling.

In summary, our results show that the loss of FN synergy site leads to normal wound closure in vivo, caused by two opposing migratory defects of activated keratinocytes in conjunction with changes in the GT that imply reduced fibrotic response due to weakened α5β1-mediated adhesions and poorly assembled FN^syn^ fibrils. In this context, peptides or antibodies that block the FN synergy site may serve as a potential antifibrotic tool in healing wounds.

## Figures and Tables

**Figure 1 cells-11-02100-f001:**
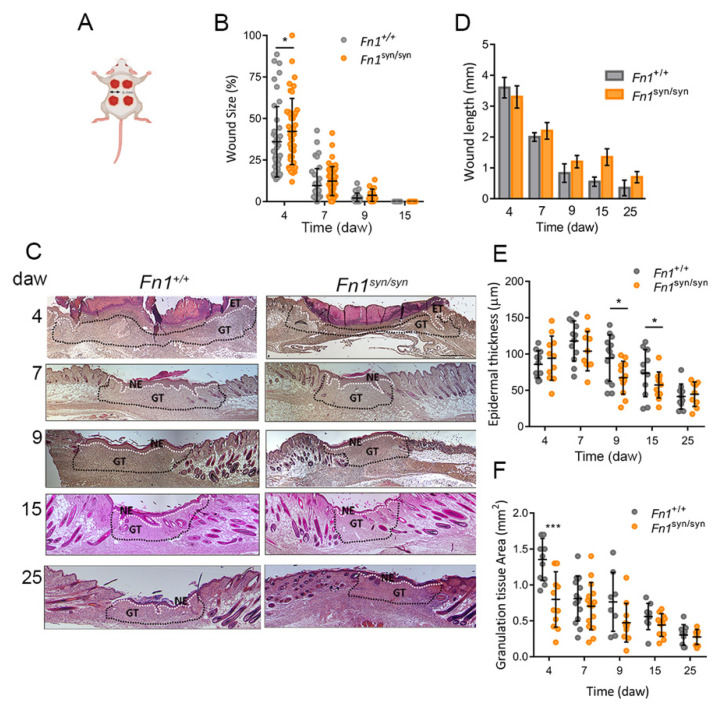
Analysis of in-vivo cutaneous wound healing in mice expressing fibronectin with the synergy site inactivated. (**A**) Schematic representation of the wound healing protocol. (**B**) Analysis of wound size calculated as the wound area at each time point referred to its area at day 0. *n* = the number of wounds analysed in *Fn1*^+/+^, *Fn1*^syn/syn^ respectively, at the mentioned time point. *n* = 57, 62 (0 days after wounding; daw); *n* = 40, 41 (4 daw); *n* = 38, 47 (7 daw); *n* = 21, 18 (9 daw); *n* = 12, 12 (15 daw). (**C**) Representative H&E images from epidermal wounds at the different time points in *Fn1*^+/+^ and *Fn1*^syn/syn^ mice. The granulation tissue is indicated with dotted lines. (**D**) Quantification of the cross-sectional wound length measured in H&E images as the distance between wound edges at the different time points. (**E**) Quantification of the mean epidermal thickness in *Fn1*^+/+^ and *Fn1^syn/syn^* wounds. (**F**) Quantification of the granulation tissue size in *Fn1*^+/+^ and *Fn1^syn/syn^* wounds. Values are shown as mean ± SEM. Distribution of the samples was assessed by Shapiro–Wilk´s test. Statistical analysis was performed using Student *t*-test, and Mann–Whitney U test was used in B. *p*-value * *p* < 0.05 and *** *p* < 0.001. ET epidermal tongue; NE new epidermis; GT granulation tissue. Scale bar in (**C**), 200 μm.

**Figure 2 cells-11-02100-f002:**
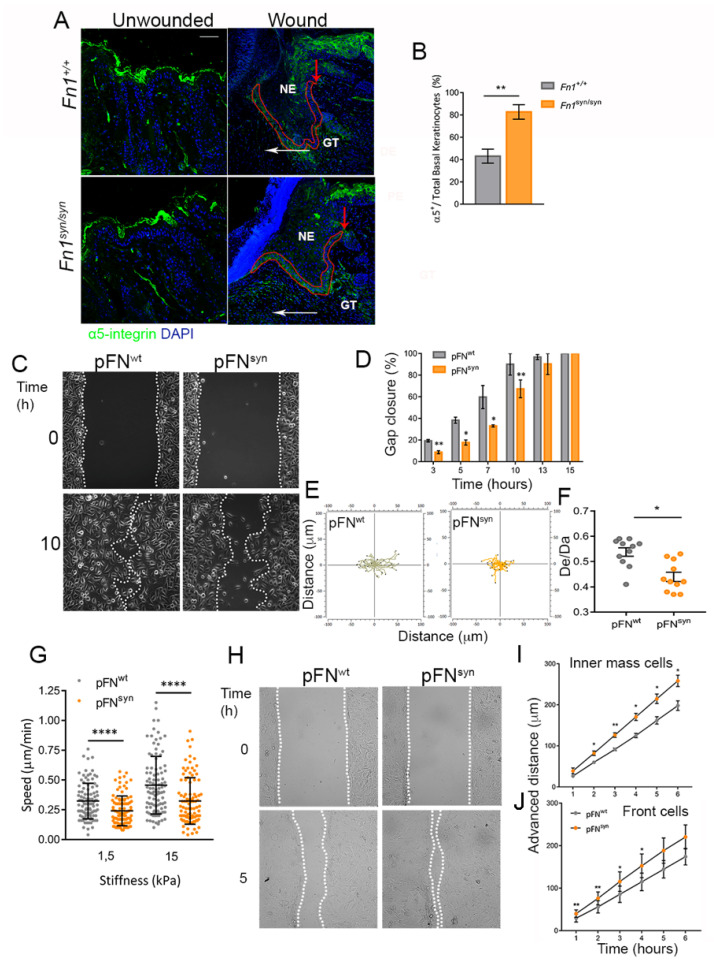
Re-epithelialization and characterization of keratinocyte migration on FN^syn^. (**A**) Representative immunofluorescence for α5- integrin (green) in unwounded skin and *Fn1^+/+^* and *Fn1*^syn/syn^ wounds at 4 daw and nuclei marked with DAPI (blue). Red arrows mark the start of the α5- integrin-expressing basal keratinocyte layer in the wounds; white arrows indicate the sense of keratinocyte tongue migration. (**B**) Percentage of α5 integrin-positive basal migratory keratinocytes referred to the total number of basal keratinocytes in the epidermal tongue (defined by a line of red dots) (*n* = 8, 6). (**C**) Images of in-vitro keratinocyte single-cell migration on pFN^wt^ and pFN^syn^ at 0 and 10 h after self-insert removal. (**D**) Percentage of gap closure calculated as the free-cell area at each time point relative to the initial area for keratinocytes migrating on pFNwt or pFNsyn at shown time points. (**E**) Representative cell tracks on pFN^wt^ and pFN^syn^. (**F**) Quantification of cell directionality in three different biological replicates, as distance Euclidean/distance accumulated (De/Da). (**G**) Analysis of cell speed of keratinocytes seeded at low density on PDMS gels with rigidities of 1.5 and 15 kPa. Three different biological replicates were analysed and around 30–50 cells were tracked in each experiment. (**H**) Representative images of keratinocyte collective migration at 0 and 5 h on pFN^wt^ and pFN^syn^. (**I**) Analysis of the cells advance in the inner mass over time. (**J**) Analysis of front cells advance over time. Four different areas were recorded per experiment and each dot represents around 8–10 cells per area for each different replicate. Results are the mean ± SEM from three independent experiments. Statistical analysis was performed using Student *t*-test or Mann–Whitney U test (**G**). *p*-value * *p* < 0.05; ** *p* < 0.01 and **** *p* < 0.0001. NE new epidermis; GT granulation tissue. Scale bar, 40 µm (**A**).

**Figure 3 cells-11-02100-f003:**
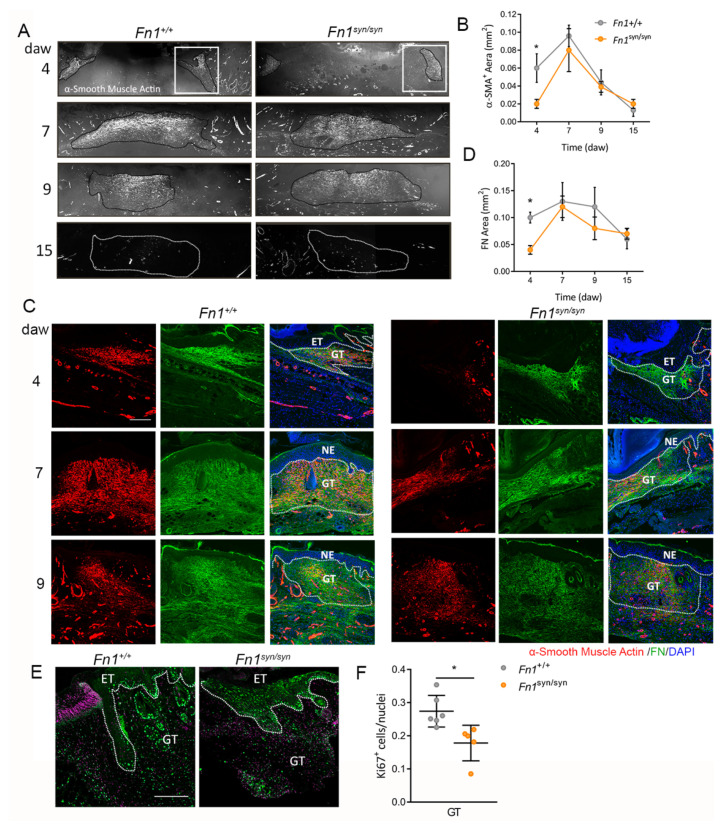
*Fn1^syn/syn^* wounds are less fibrotic. (**A**) Representative immunofluorescence of α-SMA-positive cells in the GT area (delimited by dotted line) at different time points. (**B**) Quantification of α-smooth muscle-positive areas in wounds from *Fn1*^+/+^ and *Fn1*^syn/syn^ mice. (**C**) Representative images from FN (green) distribution and α-SMA-positive cells (red) in *Fn1*^+/+^ and *Fn1*^syn/syn^ wounds at 4, 7 and 9 daw. (**D**) Quantification of FN area in the GT region of wounds from *Fn1*^+/+^ and *Fn1*^syn/syn^ mice (*n* = 12, 12 (4) *n* = 14, 12 (7) *n* = 11, 11 (9)). (**E**) Representative immunofluorescence of Ki67 immunostaining (green) and Dapi (purple) in *Fn1*^+/+^ and *Fn1*^syn/syn^ wounds at 4 daw. (**F**) Quantification of the proportion of Ki67-positive cells. Statistical analysis was performed using Student *t*-test. *p*-value * *p* < 0.05. Scale bars, 40 µm (**C**) and 200 µm (**E**). ET; epithelial tongue, GT; granulation tissue, NE, new epidermis.

**Figure 4 cells-11-02100-f004:**
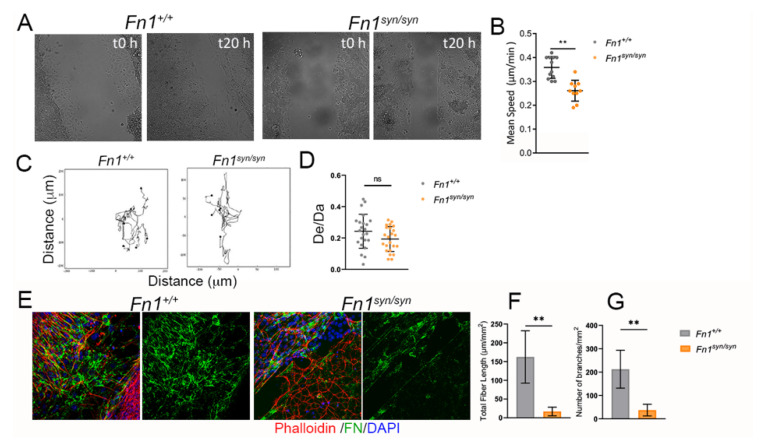
Characterization of *Fn1*^syn/syn^ fibroblast migration in scratch assays. (**A**) Representative scratch assays of *Fn1*^+/+^ or *Fn1*^syn/syn^ fibroblasts at t0 and after 20 h in serum-free medium. (**B**) Quantification of the mean speed and (**C**) analysis and (**D**) quantification of the directionality (distance Euclidean/distance accumulated; De/Da) of cells at the wound edge. Values in graphs are the mean ± SEM of 6 different leading cells from 4 different experiments. (**E**) Immunofluorescences to show FN (green) fibrils assembled by migrating fibroblasts after scratching. (**F**) Quantification of total FN fibril lengths and (**G**) number of branched FN fibrils per area. Values represent 4 images per condition. Statistically significance was determined using the Student’s *t*-test *p*-value ** *p* < 0.01.

**Figure 5 cells-11-02100-f005:**
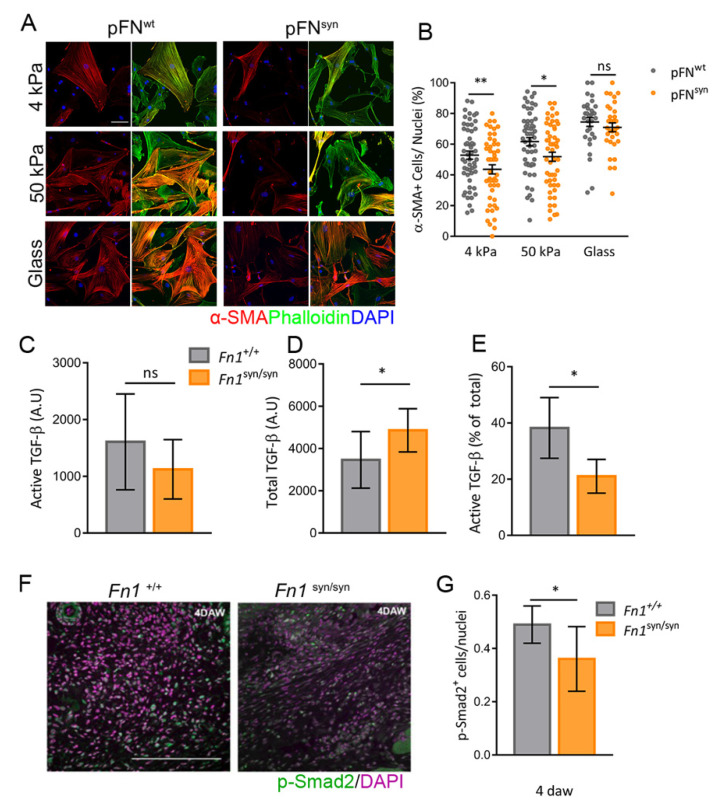
The inactivation of the FN synergy site hinders dermal fibroblast conversion to myofibroblasts. (**A**) Representative immunofluorescences of isolated DFs from *Fn1^+/+^* and *Fn1^syn/syn^* mice seeded on 4 kPa and 50 kPa PA gels and glass coverslips coated with pFN^wt^ or pFN^syn^. The actin cytoskeleton was marked with phalloidin (green), nuclei is stained with DAPI (blue) and α-SMA immunostained (red). (**B**) Quantification of the percentage of α-SMA-positive cells related to the total number of cells. Graphs show mean ± SEM of *n* = 15 different images per assay of 4 different biological replicates. (**C**) Quantification of active TFG-β1, (**D**) total soluble TFG-β1 and (**E**) ratio of active to total TFG-β1 in conditioned media from *Fn1*^+/+^ and *Fn1*^syn/syn^ DFs. Graphics show mean ± SEM of three biological assays. (**F**) Representative immunofluorescence of p-Smad2-positive cells (green) and DAPI (purple) at 4 daw in the GT. (**G**) Quantification of number p-Smad-2-positive cells per nuclei in the GT. Further, 10–12 pictures per wound were analysed from different GT areas (*n* = 7, 8 wounds, respectively). Statistical significance was determined using the Student’s *t*-test *p*-value, * *p* < 0.05 and ** *p* < 0.01. Scale bars, 20 μm (**A**) and 200 μm (**F**).

**Figure 6 cells-11-02100-f006:**
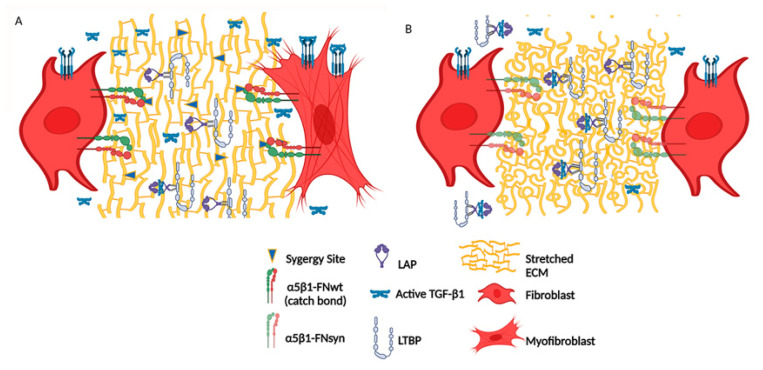
Non-proteolytic TGF-β1 activation in the granulation tissue of *Fn1*^+/+^ (**A**) and *Fn1^syn/syn^* wounds (**B**). (A) TGF-β1 is secreted non-covalently bound to the LAP-LTBP complex and is stored on FN-containing ECMs. In *Fn1*^+/+^ wounds as the ECM stiffness increases upon skin injury, FN binding cells increase traction forces on FN through synergy site catch bond formation with α5β1 integrins. TGF-β1 is activated by its liberation from the latent complex and binds TGF-β1-receptors at the fibroblast membrane inducing myofibroblast conversion. (**B**) In *Fn1^syn/syn^* wounds ECMs are less stretched trapping less TGF-β1-LAP-LTBP latent complexes, and pulling forces mediated by FN-α5β1 bonds are reduced with the consequent decrease in TGF-β1 release and signalling.

## Data Availability

The data presented in this study are available upon request to the authors.

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
