# Peer review of "The Role of the Fibronectin Synergy Site for Skin Wound Healing"

_cells, 2022, doi:10.3390/cells11132100_

Round 1

Reviewer 1 Report

The subject of the work is interesting, as it concerns the so-called synergy site which participates, together with the RGD motif, in the binding of FN to certain integrins. The problem is complex and difficult to study. The influence of the synergistic site on biological processes is rather subtle and investigating it sometimes requires really sophisticated methods. In fact, the authors have used a great number of different modern research methods. Unfortunately parts of them have been presented in a cursory and rather superficial manner, which makes it unclear for a reader not familiar with the methodology. (e.g. it would be of great help to add some explanation regarding the legitimacy of testing the presence of the Ki67 antigen in sections 3.3 or the phosphorylated Smad-2 in section 3.4, etc.).

The work contains many figures which facilitate the perception of the presented experiments. However, some corrections are needed:
Fig. 2: the meaning of arrows and abbreviations used in panel A is not explained in the legend.
Panel E is rather illegible.
Fig. 4: Is the directionality from panel D calculated in the same way as in Fig. 2F? If so, why is it presented in a different form? Perhaps it would be worth adding an explanation of how directionality was quantified?
In panel G - what does the number of branches, expressed as a fraction, mean?
Fig. S2: There is only one panel, so why mark A?
Fig. S4: There is no legend to panel A.

The use of parametric statistics throughout the work raises some doubts. One of the assumptions of parametric statistics is the normal distribution of variables. However, the data presented in Fig. 1B or in Fig. 2G does not seem to be meeting this condition. Maybe the use of nonparametric satistics would be more suitable for them?

Literature citations require correction. Reference 23 is nowhere cited in the text.
Citation adequacy check is required, e.g. reference 24 is inappropriately cited in sections 2.2 and 3.2.

I have one question for the Authors: Do you have any observations of how the FN synergy site mutation affects scarring of mice?

Author Response

Point-by-point response to reviewer’s comments

Dear Editor, dear reviewers,

First, we would like to thank you very much for your highly valuable input, comments, and suggestions in order to improve our submitted manuscript. In the following part, we would like to comment directly to the points raised by the two reviewers. In the revised manuscript, we track all the changes.

Referee 1

…. the authors have used a great number of different modern research methods. Unfortunately parts of them have been presented in a cursory and rather superficial manner, which makes it unclear for a reader not familiar with the methodology. (e.g. it would be of great help to add some explanation regarding the legitimacy of testing the presence of the Ki67 antigen in sections 3.3 or the phosphorylated Smad-2 in section 3.4, etc.).

We thank the reviewer for her/his supportive comment on our study. We revised Methods and Results sections to make clear our approaches. Upon wounding, dermal fibroblasts are activated and become proliferative and migratory to form the granulation tissue. The maximum levels of proliferation at the wound bed are followed by a high deposition of ECM proteins (Rognoni et al., 2018). Then, the late appearance of ECM proteins in Fn1syn/syn granulation tissue could be explained by an impaired activation of fibroblasts in the absence of the synergy site. For that reason, we used Ki67 proliferation marker to determine whether lower cell division could be the reason for a delay in the GT formation. We extended this explanation in section 3.3.

TGF-β is considered the master pro-fibrotic cytokine. TGF-β is secreted at the wound site and has been demonstrated to induce collagen and FN synthesis, as well as the expression of α-SMA protein in surrounding fibroblasts that become myofibroblasts. TGF-β1 contributes to myofibroblasts differentiation through both canonical and non-canonical signaling pathways (Penke and Peters-Golden, 2019). In the canonical signaling pathway, phosphorylation of Smad2/3 proteins are directly downstream TGF- β1 receptor activation. Thus, we studied the levels of phosphorylated Smad2 as marker of TGF-β1-induced cell signaling. In the revised section 3.4, we added more information and a reference about the relevance of this marker.

The work contains many figures which facilitate the perception of the presented experiments. However, some corrections are needed:

Fig. 2: the meaning of arrows and abbreviations used in panel A is not explained in the legend.

We apologize for the mistakes. We added this information

Panel E is rather illegible.

We augmented its size in the revised figure.

Fig. 4: Is the directionality from panel D calculated in the same way as in Fig. 2F? If so, why is it presented in a different form? Perhaps it would be worth adding an explanation of how directionality was quantified?

In the revised version we calculated directionality in Fig 4D in the same way as in Fig 2F, as the ratio of Euclidean distance to Accumulated distance. We also increased the number of cells analysed. Unfortunately, although the message of the results does not change, now the decrease in directionality is not significant in Fig 4D. We apologize for it and changed this point in the text of Results and explained in the figure legends.

In panel G - what does the number of branches, expressed as a fraction, mean?

We agree with the referee and thank her/his advice. We modified the graphics and expressed Fig 4F as total fiber length per area and Fig 4G as number of branches per area.

Fig. S2: There is only one panel, so why mark A?

We eliminated A

Fig. S4: There is no legend to panel A.

Done

The use of parametric statistics throughout the work raises some doubts. One of the assumptions of parametric statistics is the normal distribution of variables. However, the data presented in Fig. 1B or in Fig. 2G does not seem to be meeting this condition. Maybe the use of nonparametric statistics would be more suitable for them?

We thank the reviewer for this highly valuable advice. In the revised manuscript, we performed Shapiro-Wilk’s test to analyze data distribution, showing that, effectively, in Figs 1B and 2G it is not normal. Therefore, we applied a non-parametric Mann-Whitney U test to analyze these data. Now, in Fig 1B at 4 daw we have a significant decrease in Fn1syn/syn wounds and results in Fig 2G increased their significance.

Literature citations require correction. Reference 23 is nowhere cited in the text. Citation adequacy check is required, e.g. reference 24 is inappropriately cited in sections 2.2 and 3.2

We changed incorrect citations, added two more and checked all.

Do you have any observations of how the FN synergy site mutation affects scarring of mice?

We studied skin wounds till 25 days after wounding. At this healing stage we could not appreciate differences in ECM proteins deposition between the two genotypes in the regenerated skin nor in the remaining GT area (as is shown in Figure 1C). At this age we assume that there were no differences. However, we cannot discard that in much later stages, when all the GT has been replaced, the appearance of new skin appendages such as hair follicles or the presence of remnant collagen fibers is altered in Fn1syn/syn mice. 

Reviewer 2 Report

This study from Gimeno-Lluch et al. shows the importance of the synergy site in FN in the context of wound healing. Using appropriate in vitro and in vivo models they detail the role of this site in each stage of the wound healing process. Wounds in synergy mutant mice have reduced fibroblast content in the GT region and this reduces the wounds ability to withstand mechanical stress.

The article is well written and the conclusions drawn are appropriate from the data presented. I have a few suggestions and comments.

With regards to the Fn syn/syn mice, are the mice healthy in general with regards to size etc. Benito-Jardon et al (E life 2017) previously showed Fn syn/syn mice showed vascular issues and embryonic death. Is the Fn mutation under the control of a keratin or collagen promoter to select for keratinocytes or fibroblasts. In general more information on the generation of these mice and the basic health of the mice is required in the methods and results section.

In Figure 2A is the unwounded skin from the Fn1 +/+ or syn/syn mice. I believe unwounded skin from both mice are required for analysis in this figure.

In figure 2C Are the keratinocytes used for this experiment human/mouse and are they healthy.  

In figure 5A are the fn1+/+ and syn/syn fibroblasts paired with their equivalent fibres? If so it would be informative to repeat with an unpaired assay? 

Author Response

With regards to the Fn syn/syn mice, are the mice healthy in general with regards to size etc. Benito-Jardon et al (E life 2017) previously showed Fn syn/syn mice showed vascular issues and embryonic death. Is the Fn mutation

under the control of a keratin or collagen promoter to select for keratinocytes or fibroblasts. In general more information on the generation of these mice and the basic health of the mice is required in the methods and results section.

Fn1syn/syn mice were generated by introducing a knock-In substitution in the Fn1 gene of embryonic stem cells by homologous recombination. This is a constitutive mutation within the fibronectin gene. Therefore, the expression of FN in Fn1syn/syn mutants always contains the synergy mutation (Benito-Jardon et al. 2017). The Fn1syn/syn mice are born following a mendelian distribution when breeding heterozygous Fn1syn/+ mice. These Fn1syn/syn are also fertile, have a normal life span, acquire same size and weight and show no developmental defects when compared to their wild-type littermates as described in Benito-Jardon et al. 2017. We observed hemostasis defects in the Fn1syn/syn mice when mice were challenged by tail bleeding or by arteriolar occlusion as consequence of a defect in platelet aggregation. In the same paper, we also generated a double mutant Fn1syn/syn; intgb3-/-. The double mutants had an impaired synergy site and lack integrin b3 and suffer from vascular defects and lethality at E16.5.

Since the synergy mutation alone in mice is not lethal, we have been able to use these mice to generate dermal fibroblasts and kidney fibroblasts without needing to breed into a specific promoter or generate inducible-mutant mice. We apologize for incomplete information in the text and now added extended explanation in Methods and Results sections.

In Figure 2A is the unwounded skin from the Fn1 +/+ or syn/syn mice. I believe unwounded skin from both mice are required for analysis in this figure.

This figure was from Fn1+/+ mice. We agree with the referee and thank his/her advice. Now we added to Fig 2A a5 integrin expression in unwounded skin from both Fn1+/+ and Fn1syn/syn mice.

In figure 2C Are the keratinocytes used for this experiment human/mouse and are they healthy.

This cell line is from mouse. The keratinocytes are a cell line that was isolated from Kindlin-1fl/fl mice and spontaneously immortalized and was subcloned. We used a clone that was not transfected with Cre recombinase and thus its genotype was wild type and cells were healthy. We added this explanation in Materials section.

In figure 5A are the fn1+/+ and syn/syn fibroblasts paired with their equivalent fibres? If so it would be informative to repeat with an unpaired assay?

We performed the analysis in Fig 5A by paired assay with dermal fibroblasts isolated from Fn1+/+ and Fn1syn/syn mice and seeded onto pFNwt or pFNsyn, respectively. Unfortunately, activated fibroblasts have a high ECM secretory activity and we cannot perform an unpaired assay because myofibroblasts actively secrete FN that would mask the FN in the coating.

Round 2

Reviewer 2 Report

The authors have addressed the concerns I raised in the first review and the manuscript is now suitable for publication